# Mechanical Properties and Non-Isothermal Crystallization Kinetics of Polylactic Acid Modified by Polyacrylic Elastomers and Cellulose Nanocrystals

**DOI:** 10.3390/polym15183767

**Published:** 2023-09-14

**Authors:** Weixiao Meng, Xiaojie Zhang, Xiuli Hu, Yingchun Liu, Jimin Zhang, Xiongwei Qu, Beckry Abdel-Magid

**Affiliations:** 1Hebei Key Laboratory of Functional Polymers, School of Chemical Engineering, Hebei University of Technology, Tianjin 300130, China; mengweixiao0901@163.com (W.M.); zhangxj@hebut.edu.cn (X.Z.); huxiuli@hebut.edu.cn (X.H.); zhangjimin@hebut.edu.cn (J.Z.); 2Jinghua Plastics Industry Co. Ltd., Langfang 065800, China; jinghuasuye@163.com; 3Department of Composite Materials Engineering, Winona State University, Winona, MN 55987, USA

**Keywords:** polylactic acid, polyacrylic elastomer, cellulose nanocrystal, non-isothermal crystallization kinetics, fabrication technique

## Abstract

In this paper, a polyacrylic elastomer latex with butyl acrylate (BA) as the core and methyl methacrylate (MMA) copolymerized with glycidyl methacrylate (GMA) as the shell, named poly(BA-MMA-GMA) (PBMG), was synthesized by seeded emulsion polymerization. Cellulose nanocrystal (CNC) was dispersed in the polyacrylic latex to prepare PBMG/CNC dispersions with different CNC contents. The dried product was mixed with polylactic acid (PLA) to fabricate PLA/PBMG/CNC blends. The addition of PBMG and PBMG/CNC improved the mechanical properties of the PLA matrix. Differential scanning calorimetry (DSC) was used to investigate the non-isothermal crystallization kinetics. The Avrami equation modified by the Jeziorny, Ozawa and Mo equations was used to analyze the non-isothermal crystallization kinetics of PLA and its blends. Analysis of the crystallization halftime of non-isothermal conditions indicated that the overall rate of crystallization increased significantly at 1 wt% content of CNC. This seemed to result from the increase of nucleation density and the acceleration of segment movement in the presence of the CNC component. This phenomenon was verified by polarizing microscope observation.

## 1. Introduction

With the increase in environmental protection awareness, the demand for the use of biodegradable polymers is increasing. Polylactic acid (PLA), a biodegradable polymer derived from renewable resources [1], is considered to be one of the most competitive renewable bio-based polymers because of its excellent biocompatibility and good biodegradability [2,3]. As an environmentally friendly, biodegradable plastic, PLA shows great potential as an alternative to petroleum-based plastics in various applications. However, the wide application of PLA is limited due to some inherent shortcomings, such as low impact strength, low heat distortion temperature, and slow crystallization rate [4,5]. Therefore, recently, many researchers have been working on modification of PLA to improve these properties.

The blending method is the most simple, feasible and economical way to modify PLA [6,7,8,9]. The derivative thermogravimetric curves of poly(lactic acid)/poly(3-hydroxybutyrate) (PLA/PHB) blends have indicated single-step and two-step degradation processes for individual polymers and polymer blends, respectively. It suggested immiscibility or partial miscibility between the polymers [9]. Mazidi modified PLA with PB-g-SAN core–shell impact modifier [10]. Although toughness was improved, tensile strength and Young’s modulus decreased significantly. Zhao obtained the same results using poly(ethylene-butyl acrylate-glycidyl methacrylate) components [11]. Yeo introduced multifunctional polyhydroxybutyrate rubber copolymer into PLA to act as an effective nucleating agent to accelerate the crystallization of PLA, but the storage modulus decreased significantly [12]. In order to improve the toughness of PLA, a higher content of plasticizers or rubber fillers was required, which is often accompanied by the sacrifice of tensile strength and modulus [13,14,15,16,17]. Meanwhile, PLA is a semi-crystalline polymer and secondary crystallization during heating leads to different physical properties. Therefore, crystallization and crystallization kinetics play an important role in the manufacturing process and product performance [18]. Jin prepared ramie fiber-reinforced PLA biocomposites using non-woven ramie fibers (RF) and PLA staple fibers as raw materials [19]. In order to understand the effect of RF on the crystallization behavior of PLA, the non-isothermal crystallization kinetics of PLA and PLA/RF biocomposites were studied. The results showed that RF, as a heterogeneous nucleating agent, weakened the interaction between PLA polymer chains during the crystallization process and reduced the energy barrier of PLA. Moreover, calcium carbonate [20], saponite grafted poly(methyl methacrylate) [21], talc [22], and titanium dioxide [23] nanocomposites on the matrix of poly(lactic acid) were beneficial to PLA crystallization and attributed to the role of nucleation to facilitate the non-isothermal crystallization kinetics of PLA. Thus, to promote the crystallization performance of PLA, nanocellulose [24,25], a high-performance nanomaterial, has attracted the interest of researchers because of its adjustable aspect ratio, controllable morphology, and improved mechanical properties [26]. Nanocellulose is mainly composed of crystalline domains of cellulose, an abundantly available biopolymer, and is divided into three different types according to the degree of crystallinity and the morphology of crystalline segments. 

Cellulose nanocrystals (CNCs) or cellulose nanowhiskers (CNWs) with high crystallinity are mainly prepared by controlled acid hydrolysis [27]. CNCs, rigid rod-shaped single-crystal cellulose with diameters of 1–100 nm and lengths of 10–1000 nm [28,29,30,31,32], have attracted widespread attention due to their crystal structure, high aspect ratio and large surface area (about 150 m^2^/g) [33]. Cao improved the mechanical properties and shape memory properties of polylactic acid (PLA)/epoxidized natural rubber (ENR) thermoplastic vulcanizates (TPVs) by using CNCs as an additive [34]. Borkotoky explored the thermal degradation and crystallization behavior of PLA and PLA/CNC [35]. CNC addition improved the crystallization rate of PLA, possibly following thermal nucleation and two-dimensional discotic growth [36]. PLA and CNC were compounded by a reactive extrusion process using dicarbonyl peroxide (DCP) as the free radical initiator. PLA-D-CNC exhibited much higher crystallization rates compared to neat PLA, PLA/CNC (without DCP), and PLA/DCP. CNC promoted PLA nucleation and crystallization at high temperatures under the action of DCP [37]. Non-isothermal crystallization experiments show that low-loading CNC can act as a heterogeneous nucleating agent in the matrix, but high-loading CNC is easy to agglomerate in the PLA matrix, which hinders the migration of the PLA molecular chain and inhibits the crystallization of PLA.

Most studies have shown that plasticizers or rubber fillers can effectively improve the toughness of PLA [38,39], but they often have a non-negligible effect on their tensile strength and modulus [40]. Crystallization can improve the modulus, strength, and heat resistance of PLA [41], but usually at the expense of ductility [42]. Therefore, this article combined polyacrylic elastomer (PBMG) with a reinforced component (CNC) at different contents (1%, 2%, 3%, 4%) to modify PLA. The mechanical properties and thermal crystallization kinetics of the blends were investigated. The goal is to maintain or even improve the strength while toughening PLA.

## 2. Experimental

### 2.1. Materials

Polylactic acid (PLA) (Injection grade, M_w_ = 1 × 10^5^ g/mol) was supplied from Haizheng Biomaterials Co., Zhejiang, China. Cellulose nanocrystal (CNC) was obtained from Tianjin Mujingling Biotechnology Co., Tianjin, China. Butyl acrylate (BA), and methyl methacrylate (MMA) (analytical purity) were purchased from Tianjin Damao Chemical Reagent Factory, Tianjin, China. Potassium persulfate (K_2_S_2_O_8_, analytical purity) was obtained from Tianjin Hongyan Chemical Reagent Factory, Tianjin, China. Anionic surfactant (chemical purity) was supplied from Tianjin Reagent Factory, Tianjin, China. Glycidyl methacrylate (GMA, Analytical purity) was purchased from Tianjin Heans Biochemical Technology Co., Tianjin, China. Allyl methacrylate (ALMA, Analytical purity) was received from Tokyo Chemical Industry Co., Ltd., Tokyo, Japan. 1,4-Butanediol diacrylate (BDDA, Analytical purity) was supplied from Tianjiao Chemical Co.,Tianjin, China. Deionized water (DIW) was used for all polymerization processes.

### 2.2. Synthesis of PBMG and PBMG/CNC

The poly(BA-MMA-GMA) PBMG latex was synthesized by the same procedure developed by Qu et al. [43,44] with minor modifications using higher stirring speed and shorter time. The CNC aqueous dispersion was prepared by dispersing CNC in DIW (1/3, *w*/*w*) with magnetic stirring and ultrasonic treatment for 0.5 h, respectively. Then, the CNC dispersion was poured into a fivefold diluted PBMG latex and stirred for 1 h. The content of CNC was 1%, 2%, 3%, and 4% of PBMG latex. PBMG/CNC powders were obtained by freeze-drying under a vacuum.

### 2.3. Preparation of PLA/PBMG/CNC Blends

PLA, freeze-dried PBMG powder, and PBMG/CNC powder were dried under a vacuum oven at 6 °C for 12 h. A TE-34 twin screw extruder (Nanjing Institute of Extrusion Machinery, China) was used to prepare blends with weight ratios of 90/10/0, 90/10/1, 90/10/2, 90/10/3, 90/10/4, designated as PLA/PBMG (PLB), PLA/PBMG/CNC1 (PLBC1), PLA/PBMG/CNC2 (PLBC2), PLA/PBMG/CNC3 (PLBC3), PLA/PBMG/CNC4 (PLBC4), respectively, at a screw speed of 60 rpm and barrel temperatures of 160 °C, 165 °C, 165 °C, 165 °C, 160 °C, and 160 °C. The palletized particles were dried and molded in an injection-molding machine (JPH-30, Guangdong Hongli Machine Co., Zhaoqing, China) at 160 °C.

### 2.4. Measurement and Characterization

The monomers’ weight conversions, including instantaneous conversions and overall conversions of PBMG latexes at the seed emulsion polymerization process and their particles, were calculated and measured according to Qu et al. [45].

A field emission scanning electron microscope (FE-SEM, Nova Nano SEM 450, FEI, Lausanne, Switzerland) was used to examine the PMBG and PBMG/CNC latexes’ morphologies. The detailed procedure is described in reference 45.

Tensile tests were performed on a CMT6104 microcomputer-controlled electronic universal testing machine (Shenzhen, China) according to the GB/T1040.2-2006 standard. The size of dumbbell-shaped specimens was 150 mm ×10 mm × 4.0 mm. Impact tests were carried out on a ZH-XPL-5.5D type cantilever impact testing machine (Chengde, China) according to the GB/T1843.1-2008 standard. The specimen size was 80 mm × 10 mm × 4 mm, and the notch depth was 2 mm. Before the test, the samples were conditioned at 50 ± 5% humidity and a temperature of 23 ± 1 °C for 24 h to eliminate internal stresses. A minimum of five specimens of each sample were tested as specified in the standards [45]. DSC tests of the samples (about 5–8 mg) were conducted on a PE Diamond DSC. The samples were heated from 20 °C to 200 °C at a heating rate of 10 °C/min in a nitrogen atmosphere and kept for 3 min to eliminate the thermal history. After that, the temperature was cooled to 20 °C at a rate of 20 °C/min and then increased to 200 °C at a rate of 10 °C/min. Finally, the samples were cooled to 20 °C at the rates of 1.0 °C/min, 1.5 °C/min, two °C/min, and three °C/min, respectively.

Samples of a small rice-grain size were inserted between the two sides of the heating table and observed under an Axioskop 40 hot stage polarizing microscope (POM, Karl Zeiss, Germany). First, the samples were heated to 200 °C at the rate of 30 °C/min for 3 min to eliminate the thermal history. Then, the samples were cooled to 115 °C at the rate of 30 °C/min, and the spherulites were observed by isothermal crystallization at 115 °C.

## 3. Results and Discussion

### 3.1. Conversion and Particle Diameter Analysis of Emulsion Polymerization

The instantaneous conversion and overall conversion versus reaction time during the emulsion polymerization process are shown in Figure 1a. It can be seen that the instantaneous conversion was over 90%, and the final conversion rate was 98.82%. The overall conversion increased linearly with the reaction time at the growth stage, indicating that the drip-feeding rate of the monomers was appropriate in this reaction. Figure 1b shows the variation of theoretical and measured latex particle diameter versus reaction time. The results showed that the theoretical values were consistent with the measured ones. This indicated that all the added monomers were polymerized on the surfaces of the original latex particles without secondary nucleation, and the surfactant dosage supplemented in the pre-mixed monomers was appropriate. This also showed that during the growth stage, the particles grew under monomer-starved conditions. Under these conditions, the copolymer composition in the diameter direction would be uniform and approximately equal to the composition of the comonomer feed mixture. Similar results were obtained in a previous study by the authors [44]. The final particle diameter was 282 nm. Meanwhile, the particle diameter index (PDI) of the final latex was 0.06, meaning that all particle sizes were uniform. The same particle size and the distribution of PBMG/CNC composite latexes were obtained after the addition of CNC dispersions; i.e., there was no effect on the final PBMG latex as the CNC dispersion was added.

### 3.2. Particle Morphology and Structure

The SEM images of the PBMG and PBMG/CNC1 powders are shown in Figure 2. It can be seen in Figure 2a that the particle diameter of PBMG is consistent and evenly distributed. The particle diameter was consistent with the DLS test results. As can be seen in Figure 2b, the PBMG latex dispersed with CNC1 has a uniform particle size, indicating that CNC does not affect the morphology of PBMG. In addition, the CNC is wrapped on the surface of PBMG particles or dispersed between PBMG particles.

### 3.3. Mechanical Properties

Figure 3 shows the mechanical properties of PLA and its blends, in which PLB and PLBC represent PLA/PBMG blend and PLA/PBMG/CNC blends, respectively. It can be seen from the stress–strain curve in Figure 3a that PLA is a brittle material without a yield point. The PLA/PBMG blend shows a yield point, and the elongation at break increases from 9.80% to 17.22%. This indicates that the PLA undergoes a brittle-ductile transition, and the PLA/PBMG blend gains toughness. The notched impact strength of the PLA/PBMG blend increases from 2.60 kJ/m^2^ of PLA to 10.45 kJ/m^2^, while the tensile strength decreases from 54.23 MPa to 49.79 MPa, as shown in Figure 3a,b. Notably, the tensile strength of the blend increases and elongation at break decreases slightly after the CNC component is added. When the amount of CNC is 1 wt%, the tensile strength increases to 56.96 MPa, which is more than that of the PLA matrix. This indicates that CNC has significant reinforcing properties. From Figure 2, the aspect ratio of CNC is 14.3 ± 1.2; it can significantly improve the tensile strength of the toughened PLA system. At the same time, the impact strength (6.47 kJ/m^2^) is about 3 times higher than that of the PLA. This is attributed to the following two points. First, CNC has a higher aspect ratio, higher modulus and higher crystallinity [46]. Secondly, there are hydrogen bonds between the CNC and PLA [47]. However, the CNC powder was easy to agglomerate [48,49] when the amount of CNC added to the matrix was high, which led to uneven dispersion of the filler and a decline in the properties of the blends. Overall, Figure 3 shows that the PLA/PBMG/CNC1 had the best mechanical and toughness properties of all the blends. Therefore, the following discussion will focus on the properties of the PLA/PBMG/CNC1 blend.

### 3.4. Crystallization

Figure 4a,b show the secondary heating DSC curves and the cooling crystallization curves of the PLA blends, respectively. The glass transition temperature (*T_g_*), cold crystallization temperature (*T_cc_*), and melting temperature (*T_m_*) of the polymer can be seen in Figure 4a, and the specific values are shown in Table 1. Compared with PLA, the heating curves of PLA/PBMG and PLA/PBMG/CNC1 blends do not change significantly, which indicates that the fillers do not affect the crystallization process of PLA. At the same time, *T_cc_* and *T_m_* also move towards a lower temperature, which is attributed to the heterophase nucleation effect of the PBMG and PBMG/CNC1. This indicates that the addition of PBMG and PBMG/CNC1 promoted the movement of PLA chain segments [50] so that they have sufficient movement ability to conduct regular arrangement and crystallize at low temperatures. The lower the cold crystallization temperature, the more imperfect the crystal form would be, which leads to the melting peak moving in the low-temperature direction. Moreover, there are two different melting peaks in the PLA, which indicates that incomplete crystals are formed during crystallization [51].

The crystallinity (*X_c_*) [52] can be calculated by Equation (1):
(1)$$X_{c}=\left(\Delta H_{m}-\Delta H_{c}\right)/\left(f\times \Delta H_{m}^{0}\right)$$
where Δ*H_m_*^0^ is the melting heat of 100% crystallization of PLA, *f* is the weight percentage of PLA, where the content of powder is 10%, so *f* is 0.9. The Δ*H_m_*^0^ of pure PLA is 93.7 J/g [52]. Relevant values are summarized in Table 1. It can be seen that the addition of fillers has improved the crystallization of PLA, and more so with the PLA/PBMG/CNC1 blend.

Figure 4b shows the DSC curves of cooling crystallization at a cooling rate of 1.5 °C/min. The crystallization temperature (*T_c_*) of PLA/PBMG and PLA/PBMG/CNC1 blends are higher than that of PLA. This indicates that PBMG and PBMG/CNC1 act as nucleating agents in PLA and provide more nucleation sites to induce crystallization, which makes it possible to crystallize at a higher temperature during the cooling process.

### 3.5. Non-isothermal Crystallization Kinetics

A crystallization kinetics study was performed to investigate the effects of time, temperature, and cooling rate on crystallization behavior. The crystallization behavior of polymers is usually performed under non-isothermal conditions, so in this study, the non-isothermal crystallization kinetics of PLA and its blends were explored. Figure 5 shows the DSC curves of the PLA and its blends with a cooling rate of 1.0, 1.5, 2.0, and 3.0 °C/min, respectively. *T_p_* is the crystallization peak temperature where the crystallization rate reaches the maximum, and *T*_0_ is the temperature when crystallization starts. It can be seen from Figure 5 that both *T_p_* and *T*_0_ of PLA move to the lower temperature, and the crystallization peak becomes wider as the cooling rate increases. At a lower cooling rate, there is enough time to activate the nucleus at a higher temperature. During this time, the movement speed of the polymer segment is faster than the cooling speed, and there is enough time to complete the crystallization, so the crystallization temperature becomes higher. On the other hand, at a faster cooling rate, *T_P_* moves to a low temperature at which the mobility of molecular chains is poor and the degree of perfection of crystallization is lower. It takes more time for the polymer chains to arrange into crystal lattices, and the crystallization peak becomes wider [53].

The relative crystallinity *X*(*t*) is the ratio of the peak area from the initial crystallization temperature (*T*_0_) to any temperature (*T*) and the peak area from *T*_0_ to the end crystallization temperature (*T*_∞_) [54], which can be calculated by Equation (2).
(2)$$X(t)=\frac{\int_{T_{0}}^{T}\left(\frac{{dH}_{C}}{dT}\right)dT}{\int_{T_{0}}^{T_{\infty }}(\frac{dH_{c}}{dT})dT}$$

The temperature (*T*) can be converted into time (*t*) by Equation (3), where *T* is the temperature at *t* and *β* is the cooling rate. Then, the *X*(*t*) − *t* curve is obtained, as shown in Figure 6.
(3)$$t=/T_{0}-T//\beta$$

The half-crystallization time(*t*_1/2_) is defined as the time to complete 50% of crystallization and is an important parameter of crystallization performance. Table 2 shows the *t*_1/2_ under different cooling rates. As the cooling rate increased, the crystallization time decreased. In other words, the time to reach the same crystallinity became shorter, as shown in Figure 6. At the same cooling rate, the *t*_1/2_ of the modified PLA blends was shorter than that of PLA. Moreover, with the increase in cooling rate, the *t*_1/2_ of the PLA/PBMG/CNC1 blend continued to decrease, especially at higher cooling rates, which indicates that PBMG/CNC1 had a greater influence on the crystallization properties at higher cooling rates.

The Avrami equation [54,55], Equation (4), is commonly used in the isothermal crystallization process. The equation can be written in double logarithmic form, as shown in Equation (5).
(4)$$1-X_{t}=\exp(-Kt^{n})$$
(5)$$\mathrm{lg}\left[-\ln\left(1-X_{t}\right)\right]=\mathrm{lg}K+n\mathrm{lg}t$$
where *K* is a composite rate constant involving both nucleation and growth rate parameters; *n* is the Avrami index, a mechanism constant, which depends on the type of nucleation and growth rate parameters.

However, the non-isothermal crystallization process is more complex than the isothermal crystallization process, and it is closer to the actual fabrication process. Therefore, Jeziorny [56] extended the Avrami equation to the process of non-isothermal crystallization [57]. The modified Avrami equation is shown in Equations (6) and (7).
(6)$$\mathrm{lg}K_{c}=\frac{\mathrm{lg}K}{\beta }$$
(7)$$t_{1/2}={\left(\frac{\ln2}{K}\right)}^{1/n}$$
where *K_c_* is the modified crystallization rate constant, and *β* is the cooling rate. The graphs of lg[−ln(1 − *X*(*t*))] versus lg*t* are shown in Figure 7. They demonstrate that the polymer and the blends have similar properties, and the curves slightly deviate from linearity. This is because of secondary crystallization in the polymer and the two different crystal growth rates in the low crystalline region and the high crystalline region [35]. The intercept and slope, namely lg*K* and *n*, can be obtained by curve fitting, and *K*_c_ and *t*_1/2_ can be calculated using Equations (6) and (7), giving the results shown in Table 3. In Table 3, the value of *n* is between 2.5–3.4, i.e., it is closer to 3, which indicates that the spherulites grew in a plate shape with time [58]. As the cooling rate increased, the crystallization rate constant, *K_c_*, increased, and *t*_1/2_ decreased, indicating that the crystallization rate of PLA gradually increased. Compared with pure PLA at the same cooling rate, the *K_c_*s of PLA/PBMG and PLA/PBMG/CNC1 blends increased, and *t*_1/2_ values decreased, which shows that PBMG and PBMG/CNC1 acted as heterogeneous nucleating agents, and produced more nucleation sites in the PLA matrix, thereby accelerating the crystallization rate of PLA.

Ozawa’s equation [59] is also used to describe the non-isothermal crystallization process [60]. Ozawa considered that the linear growth rate of spherulites was a function of temperature and proposed an equation similar to the Avrami equation, as shown in Equation (8).
(8)$$\mathrm{lg}\left[-\ln\left(1-X_{t}\right)\right]=\mathrm{lg}K_{T}+m\mathrm{lg}\beta$$
where *K_T_* is the cooling function related to the crystallization rate, and *m* is the Ozawa index. Although the Ozawa model was derived from the Avrami model, the Ozawa model ignores the secondary crystallization and impact of spherulites [61], and under different cooling rates, the temperature range of polymer crystallization varies greatly, so lg[−ln(1 − *X*(*t*))] did not have a linear relationship with lg *β*, as shown in Figure 8. In addition, the physical meaning of its cooling crystallization function is not clear. Similar phenomena have been described in other articles [55,62].

Therefore, Mo [63] et al. combined the Avrami equation with the Ozawa equation and proposed a new method to analyze crystallization kinetics parameters, establishing the relationship between cooling rate and crystallization time [64], as shown in Equations (9) and (10).
(9)$$\mathrm{lg}K+n\mathrm{lg}t=\mathrm{lg}K_{T}-m\mathrm{lg}\beta$$
(10)lgβ=lgZ−algt
where *Z* = *K_T_*/*K* is the cooling rate per unit crystallization time when the system reaches a certain degree of crystallinity, and *a* (*n*/*m*) is the ratio of Avrami index *n* to Ozawa exponent *m*.

Figure 9 shows the lg*β* − lg*t* curves of PLA/PBMG and PL/PBMG/CNC1 blends at different crystallinities. It can be seen that there is a good linear relationship between lg*β* and lg*t*, which indicates that Mo’s equation can describe the non-isothermal crystallization process of PLA and its blends. The values of the ratio *a* and cooling rate per unit crystallization time *Z* can be obtained from the curves, as shown in Table 4. *Z* increases with the increase of relative crystallinity in the same system, indicating that the cooling rate required to reach a certain crystallinity in unit crystallization time is increasing.

Considering the effect of cooling rate on the non-isothermal crystallization process, Kissinger [65] proposed a theoretical model that could quantitatively determine the crystallization activation energy (ΔE) in the non-isothermal crystallization process, as shown in Equation (11).
(11)$$\frac{d[\ln(\beta /T_{P}^{2})]}{d\left(1/T_{P}\right)}=-\Delta E/R$$
where *β* is the cooling rate, *T_p_* is the temperature corresponding to the crystallization peak, and R is the gas constant (8.314 J·mol^−1^·K^−1^). As shown in Figure 10, a ln(*β*/*T_p_*^2^) − 1/*T_p_* graph with a good linear relationship was obtained. After fitting, the slope of the curve, −ΔE/R, and the calculated ΔE values are listed in Table 4. It can be observed that the |ΔE| of PLA/PBMG and PLA/PBMG/CNC1 blends is reduced to 155.80 kJ/mol and 147.57 kJ/mol, respectively, compared with the |ΔE| of PLA (156.64 kJ/mol). This indicates that the addition of PBMG or PBMG/CNC1 fillers reduced the crystallization energy barrier and increased the crystallization rate in the non-isothermal crystallization process [60], which, in turn, improved the crystallization properties of the PLA blends.

### 3.6. Polarizing Microscope

The POM images of the isothermal crystallization of PLA and its blends at 115 °C are shown in Figure 11. It can be seen from the figures that PLA crystals mainly exist in the form of spherulites with uniform size. Compared with PLA, the number of spherulites in PLA/PBMG and PLA/PBMG/CNC1 blends increased, while the size of spherulites decreased after isothermal crystallization for 3 min. The images show that the spherulites of PLA/PBMG and PLA/PBMG/CNC1 blends have covered the whole region after isothermal crystallization for 10 min. This is because the addition of PBMG and PBMG/CNC1 to the PLA matrix plays the role of a heterogeneous nucleating agent, providing more nucleation sites. The fillers accelerated the movement of the PLA segment and increased the crystallization rate of PLA, which made the crystal fully crystallized in a shorter time, which is consistent with the analysis results of the non-isothermal crystallization process. In addition, the PLA/PBMG/CNC1 blend has higher nucleation density, which indicates that the addition of CNC provides more nucleation sites [51,60].

## 4. Conclusions

In this paper, a PBMG emulsion with a uniform particle diameter was successfully synthesized by seed emulsion polymerization. CNC was dispersed in the emulsion by stirring and ultrasound, and PLA/PBMG and PLA/PBMG/CNC blends with different CNC contents were prepared by twin screw extrusion. When the content of CNC was 1%, the mechanical properties of PLA/PBMG/CNC1 blend were best, the impact strength was 300% higher than that of pure PLA, the elongation at break was 60% higher, and the tensile strength was increased by 5%. DSC tests showed that PBMG and PBMG/CNC1 improved the crystallization performance of PLA. The crystallinity of PLA/PBMG and PLA/PBMG/CNC1 blends increased from 3.92% of PLA to 12.65% and 14.05%, respectively. The Avrami equation modified by the Jeziorny and Ozawa equations, and Mo’s method were used to study the non-isothermal crystallization kinetics of PLA and its blends. It was found that, as heterogeneous nucleating agents, PBMG and PBMG/CNC1 provided more nucleation for the PLA matrix. Furthermore, the addition of CNCs acted as heterogeneous nucleating agents and increased the density of nucleating sites. The CNCs seem to weaken the force between the segments in the PLA polymer chains and accelerate the movement of the segments, thus reducing the activation energy of crystallization and accelerating the growth rate of spherulites.

## Figures and Tables

**Figure 1 polymers-15-03767-f001:**
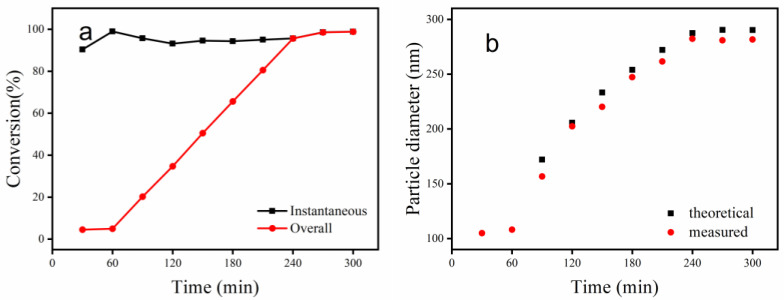
Variation with reaction time: (**a**) overall and instantaneous conversion; (**b**) measured and theoretical particle diameter.

**Figure 2 polymers-15-03767-f002:**
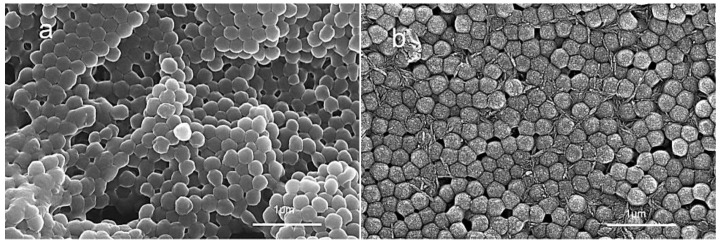
SEM images of PBMG (**a**) and PBMG/CNC1 (**b**).

**Figure 3 polymers-15-03767-f003:**
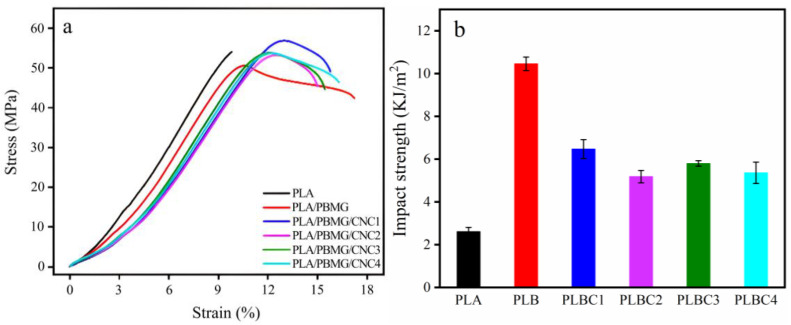
Mechanical properties of PLA blends, (**a**) stress–strain graphs, (**b**): notched impact strength of cantilever beam.

**Figure 4 polymers-15-03767-f004:**
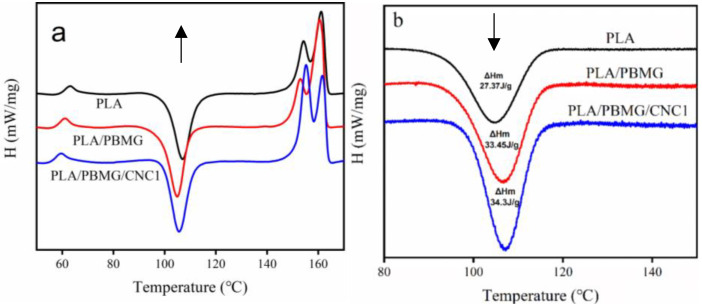
(**a**) Secondary heating curves; (**b**) cooling crystallization curves at 1.5 °C/min of PLA, PLA/PBMG, and PLA/PBMG/CNC1 blends. The arrow indicates the direction of the exothermic peak.

**Figure 5 polymers-15-03767-f005:**
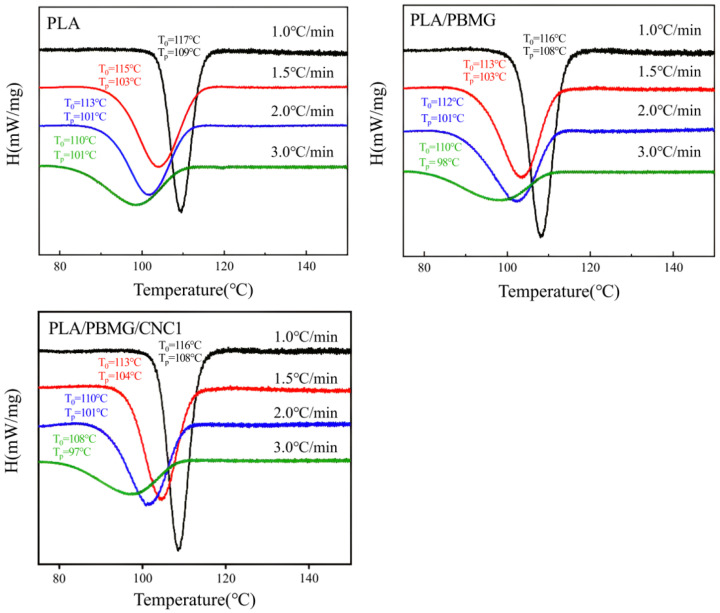
Non-isothermal crystallization curves of PLA, PLA/PBMG, and PLA/PBMG/CNC1 blends. The direction of the exothermic peak is the same as Figure 4b.

**Figure 6 polymers-15-03767-f006:**
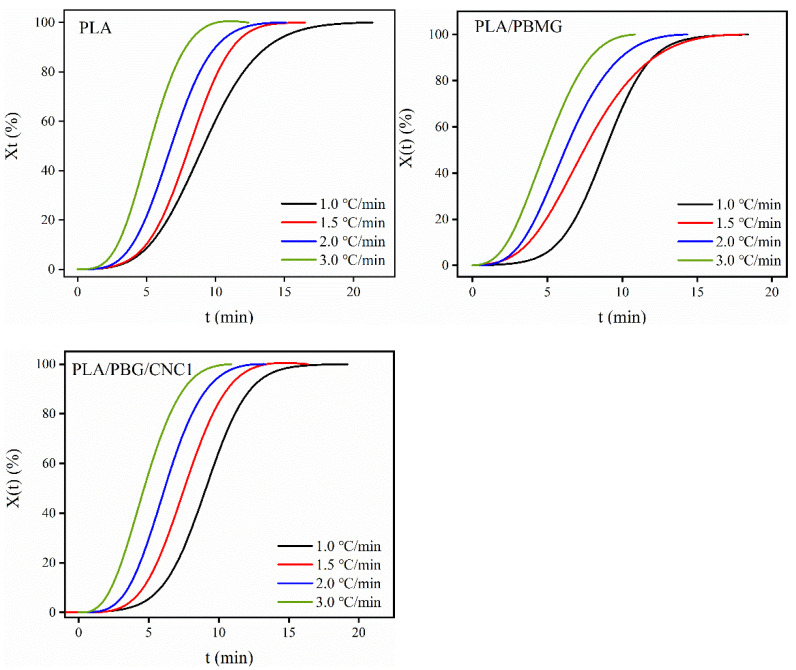
*X*(t) − *t* curves of PLA, PLA/PBMG, and PLA/PBMG/CNC1 blends.

**Figure 7 polymers-15-03767-f007:**
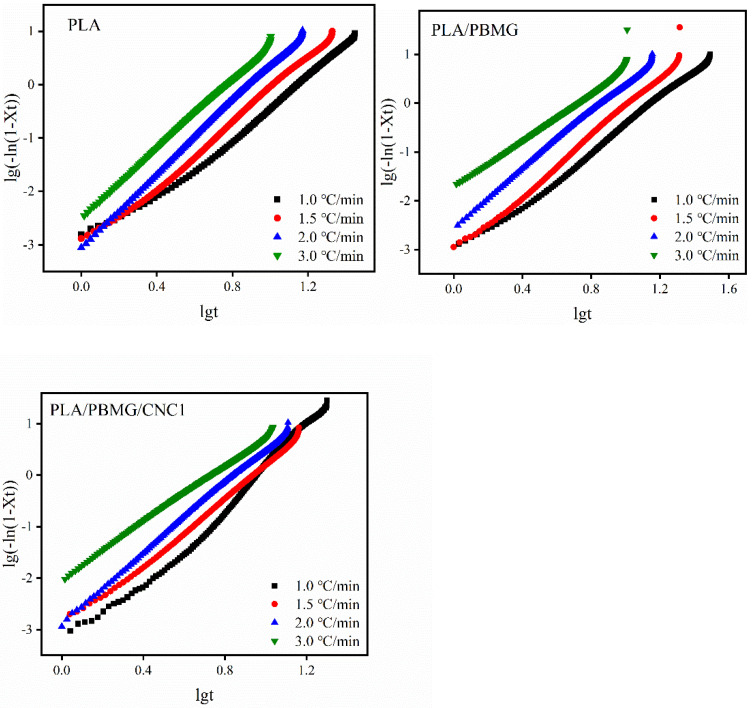
lg(−ln(1 − *X*(*t*)) − lg*t* graphs of PLA, PLA/BMG, and PLA/PBMG/CNC1 blends modified by the Jeziorny equation.

**Figure 8 polymers-15-03767-f008:**
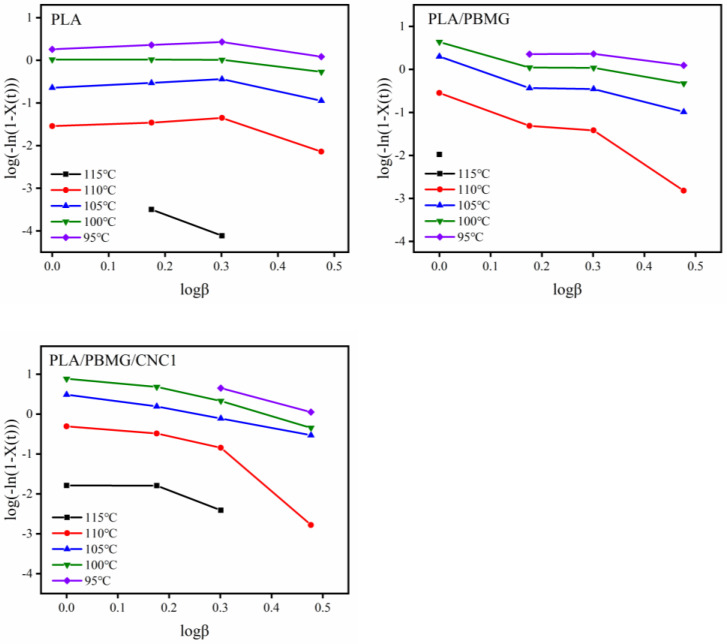
lg(−ln(1 − *X*(*t*)) − lg*β* curves of PLA, PLA/PBMG and PLA/PBMG/CNC1 blends by the Ozawa equation.

**Figure 9 polymers-15-03767-f009:**
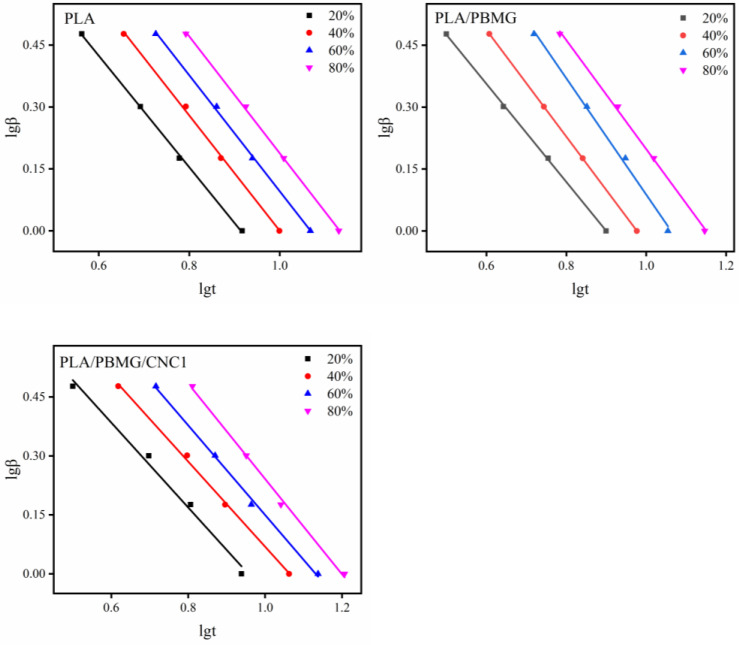
lg*β* − lg*t* curves of PLA, PLA/PBMG and PLA/PBMG/CNC1 blends at different crystallinities using Mo’s method.

**Figure 10 polymers-15-03767-f010:**
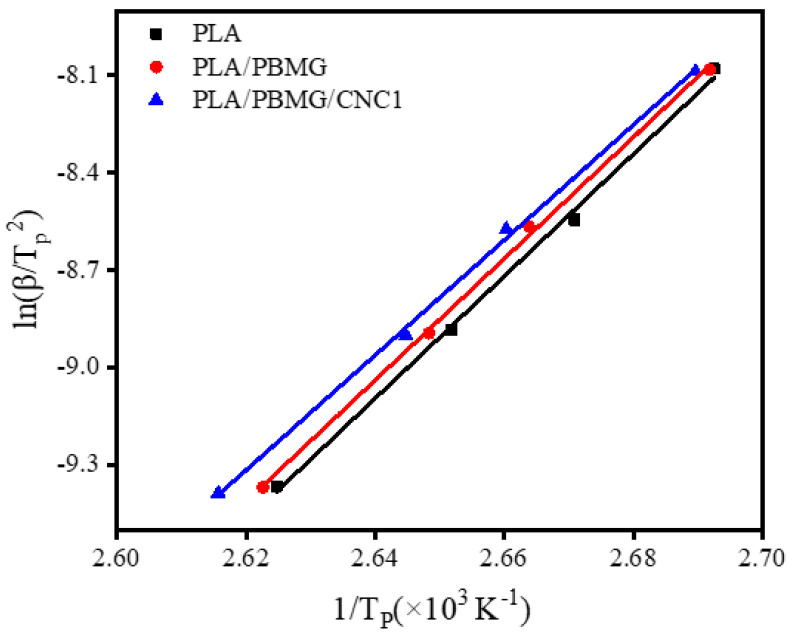
ln(*β*/*T_p_*^2^) − 1/T_P_ curves of PLA, PLA/PBMG, PLA/PBMG/CNC1 blends.

**Figure 11 polymers-15-03767-f011:**
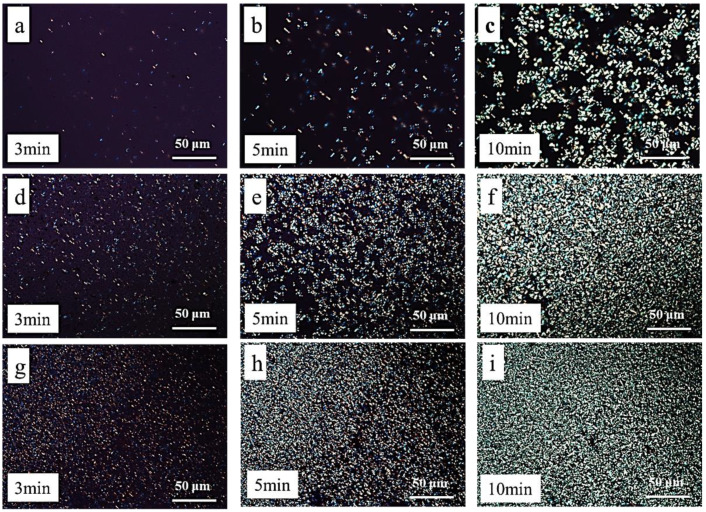
Polarizing microscopic images of isothermal crystallization with different times for PLA (**a**–**c**), PLA/PBMG (**d**–**f**) and PLA/PBMG/CNC1 (**g**–**i**) blends at 115 °C.

**Table 1 polymers-15-03767-t001:** Summary of crystallization parameters of PLA blends.

	*T_cc_* (°C)	*T_m_* (°C)	Δ*H_c_* (J/g)	Δ*H_m_* (J/g)	*X_c_* (%)
PLA	107.43	161.43	23.7	27.37	4.35
PLA/PBMG	104.70	160.30	22.78	33.45	12.65
PLB/PBMG/CNC1	105.63	155.15	22.45	34.30	14.05

**Table 2 polymers-15-03767-t002:** *t*_1/2_ values of PLA, PLA/PBMG and PLA/PBMG/CNC1 blends.

	*t*_1/2_ (min)		
	PLA	PLA/PBMG	PLA/PBMG/CNC1
1.0 °C/min	9.16	8.86	9.10
1.5 °C/min	8.14	7.48	7.50
2.0 °C/min	6.80	6.18	6.14
3.0 °C/min	5.18	4.84	4.59

**Table 3 polymers-15-03767-t003:** Non-isothermal crystallization kinetics parameters of the PLA, PLA/PBMG, and PLA/PBMG/CNC1 blends.

	*β* (°C/min)	*n*	*K*	*K_c_*	*t* _1/2_
PLA	1.0	3.20	0.0003	0.0003	11.25
	1.5	3.20	0.0006	0.0071	9.06
	2.0	3.39	0.0010	0.0317	6.88
	3.0	3.20	0.0034	0.1503	5.27
PLA/PBMG	1.0	3.15	0.0006	0.0006	9.38
	1.5	3.10	0.0008	0.0086	8.86
	2.0	2.93	0.0030	0.0547	6.41
	3.0	2.36	0.0195	0.2693	4.54
PLA/PBMG/CNC1	1.0	3.27	0.0004	0.0004	9.78
	1.5	3.32	0.0008	0.0086	7.67
	2.0	3.40	0.0013	0.0362	6.33
	3.0	2.78	0.0107	0.2205	4.48

**Table 4 polymers-15-03767-t004:** Mo’s parameters for non-isothermal crystallization kinetics of PLA, PLA/PBMG, and PLA/PBMG/CNC1 blends.

	*X* (*t*) (%)	*α*	log*Z*	*Z*	Δ*E* (KJ/mol)
PLA	20	1.18	1.1351	13.65	−156.64
	40	1.14	1.2190	16.56	
	60	1.12	1.2800	19.05	
	80	1.05	1.3030	20.09	
PLA/PBMG	20	0.96	0.9387	8.68	−155.80
	40	1.08	1.1515	14.17	
	60	1.16	1.2129	16.33	
	80	1.17	1.2849	19.27	
PLA/PBMG/CNC1	20	1.12	1.0791	11.75	−147.57
	40	1.11	1.1741	14.79	
	60	1.34	1.2166	16.47	
	80	1.54	1.2891	19.46	

## Data Availability

Not applicable.

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
