# Peer review of "Mechanical Properties and Non-Isothermal Crystallization Kinetics of Polylactic Acid Modified by Polyacrylic Elastomers and Cellulose Nanocrystals"

_polymers, 2023, doi:10.3390/polym15183767_

Round 1

Reviewer 1 Report

It was with great interest that I read the manuscript entitled "Mechanical Properties and Non-Isothermal Crystallization Kinetics of Polylactic Acid Modified by Polyacrylic Elastomer and Cellulose Nanocrystalline". In my opinion, the manuscript deals with an interesting topic that can be used in several different applications in polymer science. The used methods are adequate, the manuscript is well written and the results are supported by the results. As stated below, my only comments are related to a stronger description of the nanoparticles and the visual properties of the samples (descriptions of potential thermal degradation). Thus, I recommend this paper for publication after revisions.

Main Comments:

  1. A paper from prof. Dufresne group showed that CNC dimensions can impact polymer crystallization [1]. Please, provide a stronger description of the nanoparticles, especially its length and diameter (which can be roughly obtained in Figure 2). With this data, the authors could also estimate nanoparticles percolation threshold. This value could provide some information about composite mechanical properties. 

2. Mechanical properties are very important in this paper. Please, provide more details about the instruments and methods used in the corresponding section. I don't think that refers to another paper is enough in this particular section.

3. Those nanoparticles are also known to be sensible to thermal degradation. It would be interesting if the authors could provide information about the state of the samples after processing. Visual description of the samples, any signs of degradation, modifications in light transmittance, images of the samples, references about the expected thermal stability of the nanoparticles, etc ..

Ref. [1] - https://doi.org/10.1002/polb.24139

Other comments:

  • Authors should consider using the therm Cellulose Nanocrystals in the manuscript title.
  • Please, indicate exothermic peak direction in Figure 4.
  • In section 3.4 authors mention DMA tests. If available, authors should consider including these results as support information. If not available, authors could include a reference or remove this mention in the text.
  • Please, double-check reference numbers in the final version.

Adequate

Author Response

To Reviewer 1:

Question 1. A paper from prof. Dufresne group showed that CNC dimensions can impact polymer crystallization [1]. Please, provide a stronger description of the nanoparticles, especially its length and diameter (which can be roughly obtained in Figure 2). With this data, the authors could also estimate nanoparticles percolation threshold. This value could provide some information about composite mechanical properties.

Answer: Thank the reviewer. From Figure 2, the aspect ratio of CNC is 14.3±1.2, it can significantly improve the tensile strength of the toughened PLA system, shown in Figure 3(a). But this article only uses one kind of aspect ratio of CNC sample, so it is difficult to get the percolation threshold value for the mechanical properties. We will continue to improve our work in this area.

Question 2. Mechanical properties are very important in this paper. Please, provide more details about the instruments and methods used in the corresponding section. I don't think that refers to another paper is enough in this particular section.

Answer: The reviewer is right, and we add the instructions and methods for mechanical properties of PLA/PBMG/CNC blends used in the experiment section inserted in the revised manuscript inserted as below.

Tensile tests were performed on a CMT6104 microcomputer controlled electronic universal testing machine (Shenzhen, China) according to GB/T1040.2-2006 standard. The size of dumbbell shaped specimens was 150 mm×10 mm×4.0 mm. Impact tests were carried out on a ZH-XPL-5.5D type cantilever impact testing machine (Chengde, China) according to GB/T1843.1-2008 standard. The specimen size was 80 mm×10 mm×4 mm, and the notch depth was 2 mm. Before the test, the samples were conditioned at 50±5% humidity, and temperature of 23±1oC for 24 h to eliminate internal stresses. A minimum of five specimens of each sample were tested as specified in the standards.

Question 3. Those nanoparticles are also known to be sensible to thermal degradation. It would be interesting if the authors could provide information about the state of the samples after processing. Visual description of the samples, any signs of degradation, modifications in light transmittance, images of the samples, references about the expected thermal stability of the nanoparticles, etc ..

Answer: We appreciate the reviewer's comments, but this manuscript just discussed the preparation of a toughening agent (PBMG) for polylactic acid (PLA) and the mechanical properties of its ternary blend (PLA/PBMG/CNC) and the non-isothermal crystallization kinetics of PLA. During the preparation of the blend samples and the polarizing microscope observation, no decomposition of PBMG and CNC nanoparticles and the degradation of PLA matrix were found as the blends are milky white samples. Meanwhile, the processing conditions were chosen appropriately to avoid CNC degradation during melt processing. Therefore, we only add a sentence “The derivative thermogravimetric curves of poly(lactic acid)/poly(3-hydroxybutyrate) (PLA/PHB) blends had indicated single-step and two-step degradation processes for individual polymers and polymer blends, respectively. It suggested immiscibility or partial miscibility between the polymers [9].” in the introduction section. We will study further and contribute to another article. We hope the reviewer could understand.

Question 4. Authors should consider using the therm Cellulose Nanocrystals in the manuscript title.

Answer: I accepted your suggestion to revise the manuscript title as: Mechanical Properties and Non-Isothermal Crystallization Kinetics of Polylactic Acid Modified by Polyacrylic Elastomer and Cellulose Nanocrystal, and the expression with the same description in the whole manuscript.

Question 5. Please, indicate exothermic peak direction in Figure 4.

Answer: The heat flow arrows has been marked in Figure 4.

Question 6. In section 3.4 authors mention DMA tests. If available, authors should consider including these results as support information. If not available, authors could include a reference or remove this mention in the text.

Answer: We agree with the reviewer’s comment, and decide to delete this part shown in the revised manuscript, including in Experimental Section and Results and Discussion section.

Question 7. Please, double-check reference numbers in the final version.

Answer: We have carefully rechecked the order of the whole references shown in revised manuscript.

Reviewer 2 Report

The paper submitted by Meng et al. investigates the non-isothermal crystallization and mechanical properties of a composite based on PLA/polyacrylate copolymer and cellulose nanocrystals. The manuscript is clear, well written and the conclusions are supported by the results. The authors have proved, by using a series of techniques, the positive influence of CNC on the crystallization behaviour of the composite material. Thus, I have only minor suggestions related to the cited references which are not updated (newest reference is from 2021). In the introduction section, the authors must add some new references concerning the non-isothermal crystallization. Finally, the following reference: https://doi.org/10.1016/j.polymer.2011.05.017 may be cited in the discussion of their results. 

Author Response

To Reviewer two:

Question 1. Thus, I have only minor suggestions related to the cited references which are not updated (newest reference is from 2021). In the introduction section, the authors must add some new references concerning the non-isothermal crystallization. Finally, the following reference: https://doi.org/10.1016/j.polymer.2011.05.017 may be cited in the discussion of their results.

Answer: Thank you for your comment. We reviewed the references on PLA non-isothermal crystallization kinetics in the past two years, and it has been marked in the revised draft.

Round 2

Reviewer 1 Report

Dear Authors,

Thank you for considering my suggestions in the new version of this manuscript. In my opinion, this manuscript presents interesting data that can be useful for the community. Thus, I recommend its publication.